# Dominant-negative *STAT5B* mutations cause growth hormone insensitivity with short stature and mild immune dysregulation

Jürgen Klammt [1], David Neumann[2], Evelien F. Gevers[3,4], Shayne F. Andrew[5], I. David Schwartz[6], Denise Rockstroh[1], Roberto Colombo[7,8], Marco A. Sanchez[9], Doris Vokurkova[10], Julia Kowalczyk[4], Louise A. Metherell [4], Ron G. Rosenfeld[11], Roland Pfäffle[1], Mehul T. Dattani [12], Andrew Dauber[5] & Vivian Hwa [5]

Growth hormone (GH) insensitivity syndrome (GHIS) is a rare clinical condition in which production of insulin-like growth factor 1 is blunted and, consequently, postnatal growth impaired. Autosomal-recessive mutations in signal transducer and activator of transcription (STAT5B), the key signal transducer for GH, cause severe GHIS with additional characteristics of immune and, often fatal, pulmonary complications. Here we report dominant-negative, inactivating *STAT5B* germline mutations in patients with growth failure, eczema, and elevated IgE but without severe immune and pulmonary problems. These STAT5B missense mutants are robustly tyrosine phosphorylated upon stimulation, but are unable to nuclear localize, or fail to bind canonical STAT5B DNA response elements. Importantly, each variant retains the ability to dimerize with wild-type STAT5B, disrupting the normal transcriptional functions of wild-type STAT5B. We conclude that these STAT5B variants exert dominant-negative effects through distinct pathomechanisms, manifesting in milder clinical GHIS with general sparing of the immune system.

[1] Department of Women's and Child Health, University Hospital Leipzig, Liebigstrasse 20a, 04103 Leipzig, Germany. [2] Department of Pediatrics, Faculty of Medicine, University Hospital Hradec Kralove, Charles University, Prague, 500 05 Hradec Kralove, Czech Republic. [3] Department of Pediatric Endocrinology, Royal London Children's Hospital, Barts Health NHS Trust, Whitechapel Road, London E1 1 BB, UK. [4] Centre for Endocrinology, William Harvey Research Institute, Queen Mary University of London, First Floor North, John Vane Building, Charterhouse Square, London EC1M 6BQ, UK. [5] Division of Endocrinology, 240 Albert Sabin Way, Cincinnati Children's Hospital Medical Center, University of Cincinnati College of Medicine, Cincinnati, OH 45229, USA. [6] Mercy Kids Pediatric Endocrinology & Diabetes, Mercy Children's Hospital and Mercy Clinic, 1965 S. Fremont, Suite 260, Springfield, MO 65804, USA. [7] Institute of Clinical Biochemistry, Faculty of Medicine, Catholic University and IRCCS Policlinico Agostino Gemelli, Largo Francesco Vito 1, I-00168 Rome, Italy. [8] Center for the Study of Rare Hereditary Diseases, Niguarda Ca' Granda Metropolitan Hospital, Milan, Italy. [9] Department of Molecular Microbiology and Immunology, Oregon Health & Science University, 3181 SW Sam Jackson Park Rd, Portland, OR 97239, USA. [10] Department of Clinical Immunology and Allergology, Faculty of Medicine, University Hospital Hradec Kralove, Charles University, Prague, 500 05 Hradec Kralove, Czech Republic. [11] Department of Pediatrics, Oregon Health & Science University, Portland, OR, USA. [12] Section of Genetics and Epigenetics in Health and Disease, Genetics and Genomic Medicine Programme, University College London, Great Ormond Street Institute of Child Health, 30 Guilford Street, London WC1N 1EH, UK. These authors contributed equally: Jürgen Klammt, David Neumann, Evelien F. Gevers, Shayne F. Andrew, I. David Schwartz. Correspondence and requests for materials should be addressed to V.H. (email: Vivian.Hwa@cchmc.org)

Patients diagnosed with growth hormone insensitivity syndrome (GHIS) share common clinical characteristics of impaired postnatal growth due to low or undetectable serum IGF1 concentrations despite normal or elevated growth hormone (GH) concentrations[1–3]. Autosomal-recessive (AR) mutations of the GH receptor, GHR ("Laron syndrome", MIM 262500) are the most prevalent molecular cause of GHIS, although a few autosomal-dominant (AD) cases have also been reported (MIM 604271)[4–9]. STAT5B deficiency (MIM 245590), a rare cause of GHIS with immunodeficiency, is an AR disorder, first described in a patient with severe short stature (height SDS −7.5), who was T-lymphopenic[10, 11] and succumbed to a progressive pulmonary disease[12]. STAT5B, typical of STAT proteins, is composed of discrete protein modules including a 4-alpha helix coiled-coiled domain (CCD), a DNA-binding domain (DBD), an SH2 (src-homology 2) domain for docking to phosphorylated tyrosines, and a C-terminal transcriptional activation domain (TAD). All seven recessively inherited inactivating STAT5B mutations characterized to date lack functional SH2 and downstream TAD domains, and often the entire protein is immunologically undetectable[13]. One copy of wild-type (WT) STAT5B allele appears to be sufficient for normality as heterozygous relatives of affected patients are of normal height and without immunological or pulmonary complications[14]. Since STAT5B functions as a dimer when activated, it is conceivable that natural heterozygous STAT5B variants exist which disrupt dimeric functions. Recurrent somatic activating heterozygous missense STAT5B mutations in the SH2 or TAD domains, for example, were recently identified and reported to be causal of lymphomas[15–17]. Germline heterozygous STAT5B variants associated with impaired human growth and/or immunity have yet to be identified. Our previous functional evaluations of two rare heterozygous STAT5B missense variants identified in children with idiopathic short stature had demonstrated that the variants were unlikely to be the sole cause of growth failure[18,19].

We now report the first germline heterozygous STAT5B variants with dominant-negative effects, identified by targeted and whole-exome sequencing (WES), in short-statured subjects from three unrelated families. Neither the index patients nor affected relatives suffer from severe immunological disturbances. The three missense mutations retain the capability to become robustly tyrosine phosphorylated upon GH stimulation and, subsequently, to form dimers with themselves or with the STAT5B WT protein. However, their capacity to act as a transcription factor is blunted since nuclear import is abrogated in STAT5B with a mutation mapping to the CCD domain (p.Gln177Pro) while STAT5B proteins with DBD mutations fail to bind canonical STAT5B DNA response elements (p.Gln474Arg, p.Ala478Val). Importantly, each mutant STAT5B protein interferes with the normal functions of the WT isoform. Altogether, we demonstrate that specific heterozygous STAT5B germline mutations exert dominant-negative effects resulting in STAT5B deficiency clinically characterized by significant postnatal growth impairment, mild GH insensitivity, eczema, and elevated IgE.

## Results

### Patients and variant identification

The clinical profiles of male index patients from families 1 and 2 were consistent with GHIS, with postnatal growth failure, serum IGF1 concentrations close to (Proband 1) or below the detection limit (Proband 2), whereas basal and stimulated GH serum concentrations were normal (Table 1). For Proband 1, low serum concentrations of acid-labile subunit (IGFALS) corroborated the diagnosis of GHIS[13]; for Proband 2, the lack of response to exogenous GH in an extended stepwise IGF1 generation test (Supplementary Table 1) confirmed

a state of GHIS. Moreover, affected siblings (Fig. 1a; the monozygotic twin in family 1 and a brother in family 2) presented with comparable biochemistries and growth profiles (see "Detailed patient reports" in Supplementary Note 1 and Supplementary Table 2). Initial targeted sequencing of key genes along the GH–IGF1 axis revealed a de novo heterozygous STAT5B variant in the twin brothers of family 1 (c.530A > C, exon 5, p.Gln177Pro; CCD, Fig. 1b) and a maternally inherited c.1433C > T variant (exon 12, p.Ala478Val, DBD) in family 2. Subsequent WES analysis, performed in both families, excluded additional pathogenic variants (for criteria used in the WES analysis pipeline, see Supplementary Note 1 and Supplementary Tables 3 and 4).

For Proband 3, family 3, the top candidate variant from WES analysis was a heterozygous STAT5B c.1421A > G variant (exon 12, p.Gln474Arg; DBD). A novel heterozygous JAK2 variant (c.2374C > T, p.Pro792Ser) was also identified but did not segregate with the clinical phenotype of the family (Supplementary Note 1 and Supplementary Table 4). Proband 3 had familial short stature accompanied by autoimmune thyroiditis, celiac disease, and poor growth response to rhGH treatment (Table 1). Interestingly, Proband 3 and her two siblings who had short stature and carried the variant, but not the affected father, also presented with microcephaly (Supplementary Note 1 and Supplementary Table 2). Re-analysis of the exome data for potential variants that might contribute to the microcephalic phenotype, however, were unrevealing. Of note, probands from families 1 and 2 were not microcephalic.

None of our patients presented with symptoms of severe immune dysfunction normally associated with STAT5B deficiency[20], although the majority of carriers of the identified STAT5B variants had eczema (Fig. 1a), Proband 3 had autoimmune thyroiditis and celiac disease which were successfully controlled, and Proband 1 had childhood bronchial asthma. Immunological evaluations performed for STAT5B variant carriers in the three families were normal, with the exception of elevated IgE concentrations in eight out of nine patients (Supplementary Table 2 and Supplementary Data 1).

Detailed patient phenotype and genetic analyses reports, and immunological profiles of the three families can be found in the Supplementary Information. These are the only three families in whom a dominant-negative STAT5B mutation has been identified. Over the past decade, our research groups have investigated a total of 164 children with marked short stature and overlapping phenotypes of GHIS accompanied by variable symptoms suggestive of immune dysregulation or in whom initial GHR sequencing did not reveal any pathogenic genomic aberration.

### Mutated STAT5B proteins are phosphorylated and can dimerize

Each of the three non-synonymous STAT5B variants is private and not listed in the large-scale variant databases, with amino acid substitutions predicted to be pathogenic (Supplementary Table 5). To evaluate functional pathogenicity, N-terminally tagged STAT5B p.Gln177Pro, p.Ala478Val, and p.Gln474Arg variants were re-generated. Expression of each variant and GH-induced Tyr-phosphorylation (pSTAT5) were shown to be comparable to those of tagged WT STAT5B in reconstituted HEK293(hGHR) systems (Fig. 2a, upper panels). Co-immunoprecipitation (co-IP) experiments, furthermore, supported homo-dimerization capabilities for each variant, and, mimicking a heterozygous state, ability to hetero-dimerize with WT STAT5B (Fig. 2a, bottom panels). Interestingly, p.Gln177Pro demonstrated reproducible and robust, GH-induced phosphorylation which was markedly and time-dependently sustained (Fig. 2b). This enhanced STAT5 phosphorylation was corroborated in primary dermal fibroblasts (Fig. 2c, d) stimulated with

**Table 1 Clinical characteristics of autosomal-dominant STAT5B-deficient index patients and comparison to previously reported autosomal-recessive STAT5B cases**

|  | Proband 1 [p.Gln177Pro] | Proband 2 [p.Ala478Val] | Proband 3 [p.Gln474Arg] | Published AR cases {n} |
|---|---|---|---|---|
| Sex | Male | Male | Female | f/m = 7/3 {10} |
| Age (years) | 14.5 | 1.8 | 12.8[a] | 1.9–18.0 {10} |
| *Birth data* |  |  |  |  |
| Gestational age (weeks) | 36 | 39 | 39 | Preterm: 6/8 {8} |
| Birth weight (g) [SDS] | 2500 [−0.9] | 3460 [0.1] | 3317 [0.2] | [−2.4 to 3.0][b] {6} |
| Birth length (cm) [SDS] | 45 [−1.7] | nd | 48 [−0.2] | [−2.4 to 2.3][b] {5} |
| *Auxological features* |  |  |  |  |
| Weight (kg) [SDS] | 28.0 [−4.5] | 9.5 [−2.3] | 22.8 [−4.7] | [−6.7 to −3.6] {4} |
| Height (cm) [SDS] | 131.5 [−5.3] | 76.8 [−2.9] | 123.8 [−4.5] | [−9.9 to −4.3] {10} |
| Target height (SDS) | −0.74 | −0.83 | −1.01 | −1.98 to −0.11 {8} |
| Head circumference (SDS) | −0.53 | −1.70 | −3.73 | −1.40, −2.86 {2} |
| Bone age (years) | 9.6 | nd | 8.8 | Delayed: 7/7 {7} |
| Puberty | Delayed | na | Delayed | Delayed: 6/7 {7} |
| *Endocrine features* |  |  |  |  |
| GH, basal (ng ml$^{-1}$) | 0.4 | 3.2 | 2.0 | 0.1–17.6 {10} |
| GH, stimulated (ng ml$^{-1}$) | 16.2 | 17.3 | 4.0 | 6.6–53.8 {7} |
| IGF1 (ng ml$^{-1}$) [SDS/*reference range*] | 56 [*76–499*] | <25 [*51–303*] | 208 [−1.5] | <normal: 10/10 {10} |
| IGFBP3 (mg l$^{-1}$) [SDS/*reference range*] | 2.33[c] [−1.7] | 1.29 [*0.8–3.9*] | 3.80 [*3.9–9.4*] | <normal: 10/10 {10} |
| IGFALS [*reference range*] | 418 p mol l$^{-1}$ [c] [*986–1678*] | nd | 13 mg l$^{-1}$ [*5.6–16.0*] | <normal: 6/6 {6} |
| Prolactin (mU l$^{-1}$) [*reference range*] | 291[c] [*86–324*] | 553 [*163–1039*] | 621[d] [*<383*] | >normal: 6/7 {7} |
| *Immunological and pulmonary phenotype* |  |  |  |  |
| IgE (kU l$^{-1}$) [*reference range*] | 156/340[e] [*<200/<114*] | 118 [*<52*] | 127 [*<629*] | >normal: 4/7 {5} |
| Hemorrhagic Varicella | No | No | No | 5/5 {5} |
| Chronic pulmonary disease | Recurrent infections | No | No | 8/10 {10} |
| Lung fibrosis | nd | nd | No | 6/7 {7} |
| Lymphocytic interstitial pneumonia (LIP) | nd | nd | No | 7/8 {8}[f] |
| Eczema/skin pathology | Yes | Yes | No | 8/8 {8} |
| Otherwise disturbed immunological profiles | No | No | No | 5/7 {7} |
| Autoimmune disease | No | No | Thyroiditis; celiac disease | 6/10 {10}[f] |

nd not determined, na not applicable. Standard deviation scores (SDS) and reference ranges are shown in brackets with reference ranges in italics
[a] Patient on GH treatment for 3 months
[b] Six of eight patients born AGA (appropriate for gestational age)
[c] Determined at age 16.5 years during rhIGF treatment
[d] Measured at 15.3 years while on rhIGF1 for 4 months
[e] Measured at two occasions
[f] Diagnosis confirmed or suspected

GH or interferon gamma (IFNγ). Enhanced phosphorylation (Fig. 2d) was not due to changes in basal *STAT5B* or *GHR* expression patterns which were similar to those in control fibroblasts (Fig. 2c, e).

**Nuclear import of activated STAT5B p.Gln177Pro is abrogated**. We next asked whether all phosphorylated STAT5B species could translocate to the nucleus, a necessary step for transcriptional actions. Immunofluorescent staining and deconvolution microscopy revealed that the tagged p.Gln177Pro variant, despite robust GH-induced phosphorylation and in contrast to WT and the other two STAT5B variants, failed to enter the nucleus, remaining localized to the cytoplasm and accumulated at the nuclear membrane (Fig. 3a). Strikingly, when co-expressed with WT STAT5B, not only did the p.Gln177Pro variant remain cytoplasmic upon GH stimulation, but co-localization of the variant and WT STAT5B was detected only in the cytoplasm and by the nuclear membrane (Fig. 3b, bottom panels). Our results strongly suggest that the p.Gln177Pro variant was defective in nuclear localization and, when dimerized with WT STAT5B, prevented the WT from mobilizing into the nucleus. Conversely,

WT STAT5B could not facilitate the nuclear localization of associated p.Gln177Pro, suggesting that motifs or regions within each of the associated monomers may both be required for appropriate nuclear import.

**STAT5B p.Gln474Arg and p.Ala478Val are DNA-binding defective**. For the p.Gln474Arg and p.Ala478Val variants, located within the DBD module, we evaluated whether DNA binding might be impaired, employing standard gel-shift electrophoretic mobility shift assay (EMSA) analysis. WT STAT5B readily bound and gel-shifted the STAT5B-specific DNA probe (GHRE) only under GH-stimulated conditions (Fig. 3c). However, neither of the two variants could gel-shift the probe, clearly demonstrating loss of DNA-binding functions. When Myc-tagged STAT5B variants were co-expressed with WT FLAG-STAT5B, gel-shifting was observed under GH-stimulated conditions (Fig. 3d), but this was attributed to bound WT FLAG-STAT5B, as the complexes were further shifted (supershifted) by anti-FLAG antibody and not by anti-Myc antibody (Fig. 3d). These results were consistent with the absence of p.Gln474Arg or p.Ala478Val peptides from the GHRE probe/protein complexes, suggesting that the protein

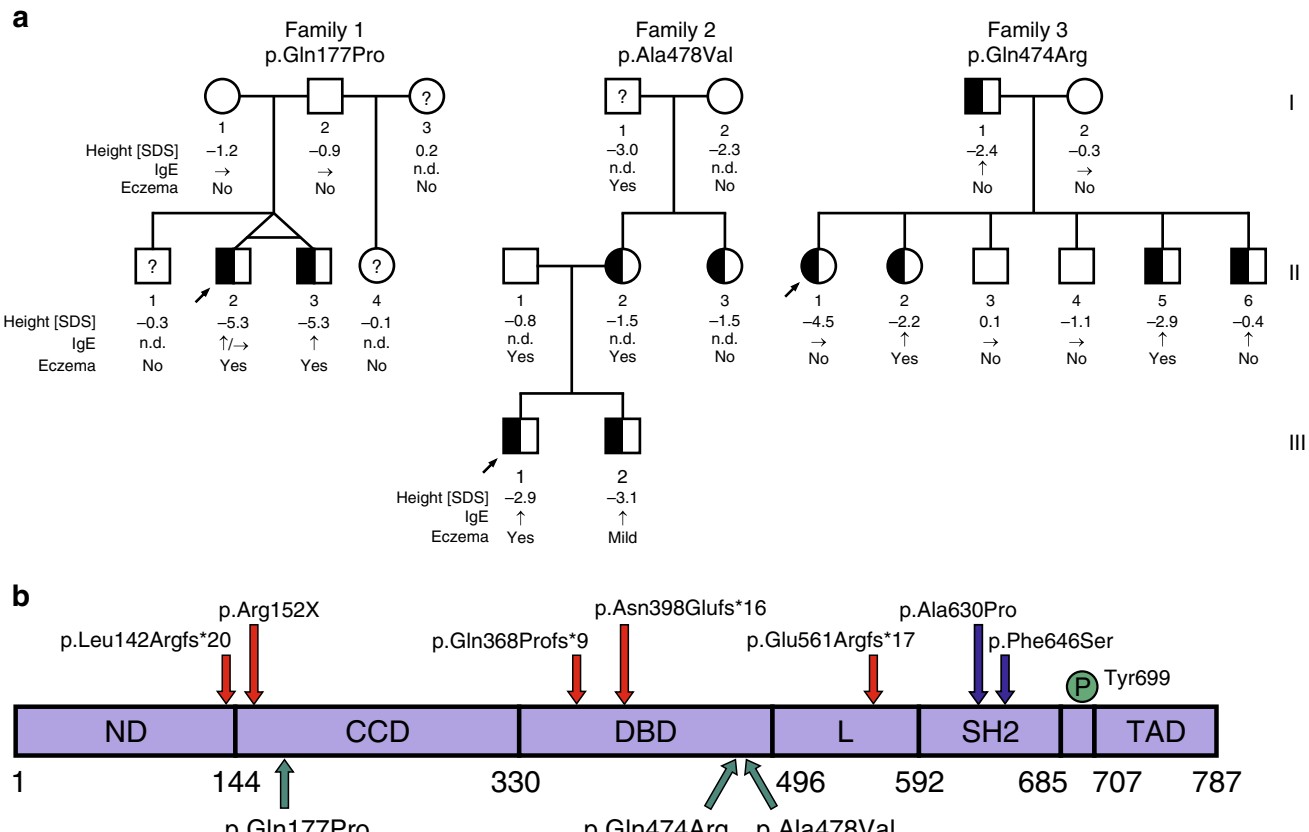

**Fig. 1** Segregation of heterozygous STAT5B missense mutations in the families. **a** Individuals bearing de novo (family 1) or inherited (families 2 and 3) heterozygous *STAT5B* mutations are indicated by half-filled symbols. Standard deviation score of last reported height [SDS], relative immunoglobulin E [IgE] levels, and the occurrence of eczema are shown below the symbols. Index patients are marked by an arrow. Question marks indicate an unknown genotype. n.d., no data available. **b** Schematic of the STAT5B protein (drawn to scale) with major functional domains (domains: ND, N-terminal; CCD, coiled-coil; DBD, DNA-binding; L, linker; SH2, src-homology 2; TAD, transactivation), their boundaries, and tyrosine residue (Tyr699) whose phosphorylation is necessary for STAT5B activation. Positions of previously reported homozygous amino acid substitutions are shown above the bar (arrows: red, protein truncating variants; blue, missense); heterozygous missense mutations identified in this study are indicated below the bar (green arrows)

component in all shifted complexes was composed solely of phosphorylated, homo-dimeric, WT STAT5B.

The failure of STAT5B variants p.Gln474Arg and p.Ala478Val to bind GHRE parallels the inability to drive transcriptional activities as demonstrated by STAT5B-mediated luciferase reporter assays. Neither of the two variants by themselves could drive expression of the luciferase reporter (Fig. 3e) and when co-expressed with WT STAT5B, the transcriptional activities of WT STAT5B were significantly blunted compared to WT STAT5B alone. Altogether, the results provide strong evidence that these DBD variants ablated STAT5B DNA-binding capabilities and prevented associated WT STAT5B from binding DNA and functioning as a transcription factor.

## Discussion

To date, the clinical condition of STAT5B deficiency has been described exclusively as an AR trait, with the majority of mutations causing early protein termination (nonsense, frameshift mutations) and total STAT5B deficiency. Only two homozygous STAT5B missense mutations have been reported, both mapping to the SH2 domain and affecting overall protein stability and/or functions[10,21] (Fig. 1b). Here we report the identification of three novel germline STAT5B missense variants, with demonstrable dominant-negative effects, associated with short stature and mild GHIS in three unrelated families. The two loss-of-function

missense variants in the DBD module are four amino acids apart, while p.Gln177Pro is located in the CCD module (Supplementary Fig. 1a, b). Each of the variants interacted with WT STAT5B, exerting dominant-negative effects that, ultimately, reduced transcriptional activity.

The p.Gln177Pro substitution, located towards the C-terminal end of the first α-helix in the CCD (Supplementary Fig. 1e), ablated ability to nuclear localize despite robust expression and GH-induced phosphorylation. The predicted disruption of α-helix 1 by the proline substitution, surprisingly, did not destabilize the whole protein (unlike the STAT5B p.Ala630Pro[10]). However, the functional integrity of the four α-helices that comprise the CCD was likely disrupted, as the helix bundle was recently shown to act in concert as an unconventional nuclear localization signal[22]. The inability of phosphorylated p.Gln177Pro to translocate to the nucleus, where resident nuclear phosphatases act on phosphorylated STAT proteins[23,24], could explain the accumulation of cytosolic, phosphorylated STAT5B species. Interestingly, impaired nuclear dephosphorylation also has been reported for AD, gain-of-function, STAT1 CCD mutants that are associated with chronic mucocutaneous candidiasis (MIM 614162), but none were reported as defective in nuclear localization[23,25]. CCD loss-of-function missense mutations in STAT3 (ref. [26]), STAT2 (ref. [27]), and STAT6 (ref. [28]) have yet to be reported. Hence STAT5B p. Gln177Pro is the first described STAT variant with a nuclear localization defect and distinct dominant-negative actions.

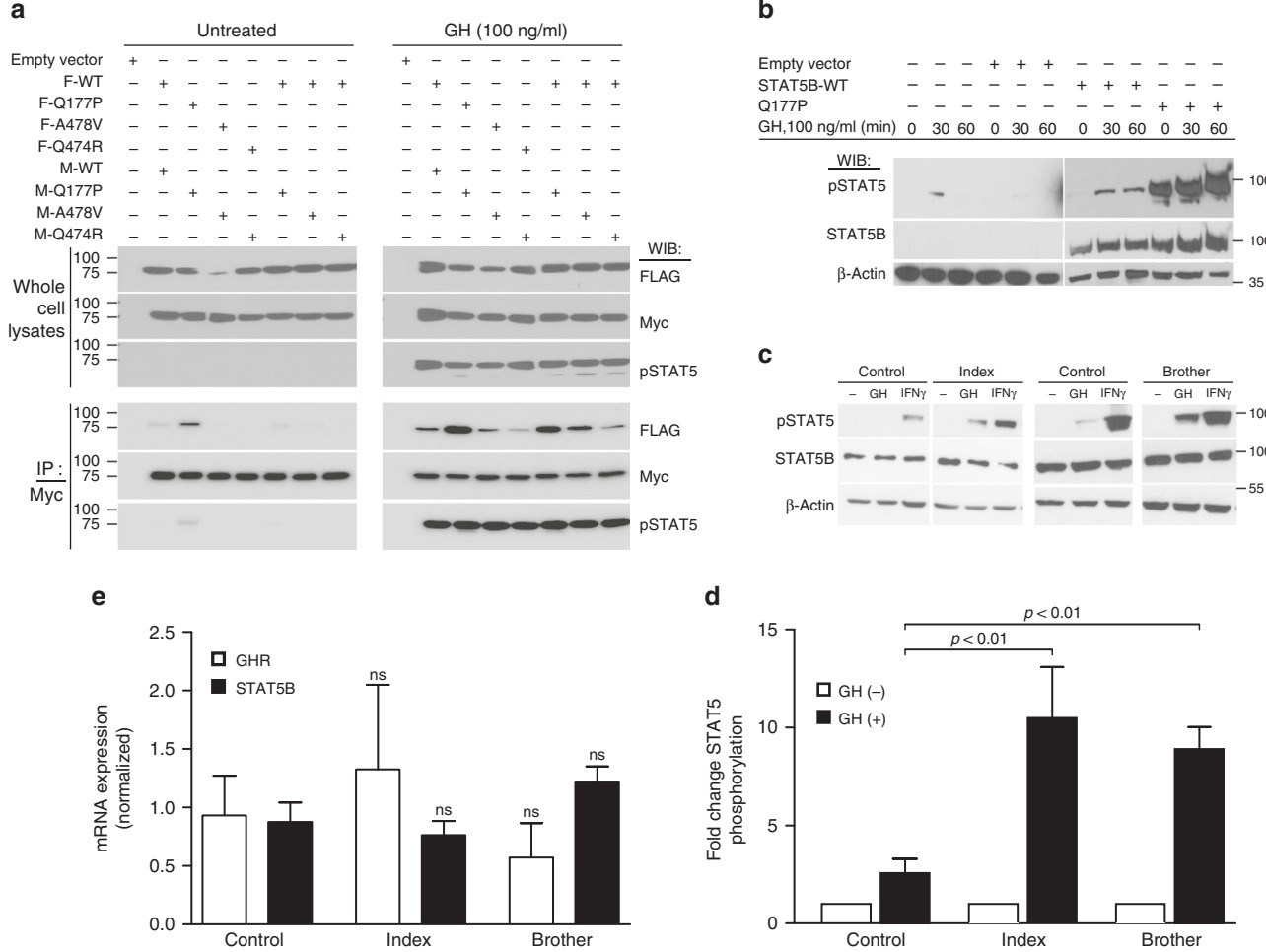

**Fig. 2** GH-dependent STAT5B activation is retained by the missense variants. Whole-cell lysates from untreated or GH-stimulated (30 min or as shown) HEK293(hGHR) (**a**) or HEK293 (**b**) cells transfected with the indicated FLAG- (F) or Myc-tagged (M) STAT5B wild-type (WT) or variant plasmids (p. Gln177Pro, Q177P; p.Ala478Val, A478V; p.Gln474R, Q474R) were subjected to immunoprecipitation (IP) and/or immunoblotting (WIB) using antibodies as indicated (pSTAT5, phospho-Tyr-STAT5). Representative immunoblots of three experiments are shown. **c–e** Whole-cell lysates were prepared from primary dermal fibroblasts derived from affected twins in Family 1 and control fibroblasts, after stimulation with GH or IFNγ and immunoblotted for pSTAT5, total STAT5B, or β-actin as a loading control. Representative blot out of four independent experiments is shown. **d** Densitometric evaluation of pSTAT5 determined in patients' fibroblasts (fold change compared to untreated cells). pSTAT5 levels were normalized to total STAT5B and β-actin protein amounts in each sample. Data are shown as mean ± S.E.M. of three independent experiments. **e** qPCR verification of *STAT5B* and *GHR* mRNA expression determined in fibroblasts from the index patient, his brother, and control cells. Mean ± S.E.M. of at least three independent experiments is shown. Statistical analysis was performed by one-way ANOVA followed by Tukey's post hoc test, *p*-values relative to control are indicated (ns, not significant)

The normal phosphorylation and nuclear trafficking of our dominant-negative STAT5B DBD mutants, p.Gln474Arg and p. Ala478Val, is remarkably similar to characterized, dominant-negative STAT3 DBD mutations[29–31] associated with hyper-IgE syndrome (MIM 147060)[31,32]. The affected STAT5B residues, Gln474 and Ala478, are conserved among all members of the STAT family (Supplementary Fig. 1a). Gln474 resides within, and Ala478 close to, a sequence segment that occupies the major grove of bound DNA in all previously investigated STAT homologous sequences (STAT1 (ref. [33]), STAT3 (ref. [34]), STAT6 (ref. [35]); Supplementary Fig. 1b–d). The STAT5B p.Gln474Arg, in particular, would be predicted to directly interfere with DNA response element recognition, possibly by disrupting direct hydrophobic and polar contacts between glutamine and thymines and phosphates on the DNA backbone[34], as was suggested for the analogous heterozygous STAT3 p.Gln469His and p.Gln469Arg mutations (Supplementary Fig. 1c, d). Altogether, heterozygous expressed missense mutations within the STAT DBD domains

can negatively impact normal dimeric STAT transcriptional functions, resulting in clinical syndromes and disorders.

All three heterozygous loss-of-function STAT5B missense variants lead to postnatal growth restriction for the index patients (height ranged from −2.9 to −5.3 SDS). Growth impairment in our patients is likely due to the consequence of diminished IGF1 serum concentrations and, presumably, extrahepatic resistance to GH, e.g. at the growth plate[36]. Intra- and interfamilial variability of stature in the three families was comparable to that normally seen in AR GHIS[1]. Strikingly, the growth deficit was in the range of patients carrying dominant-negative *GHR* mutations[4–9] (Supplementary Fig. 2), although heights were above −2 SDS in 3 out of the 11 individuals carrying dominant *STAT5B* mutations, one of whom, II-6 of family 3, is quite young and will be closely monitored. Overall, however, the phenotypes of our patients are clearly distinct from those of heterozygous relatives of AR STAT5B-deficient patients who typically have heights within the low normal range[14] (Supplementary Fig. 2).

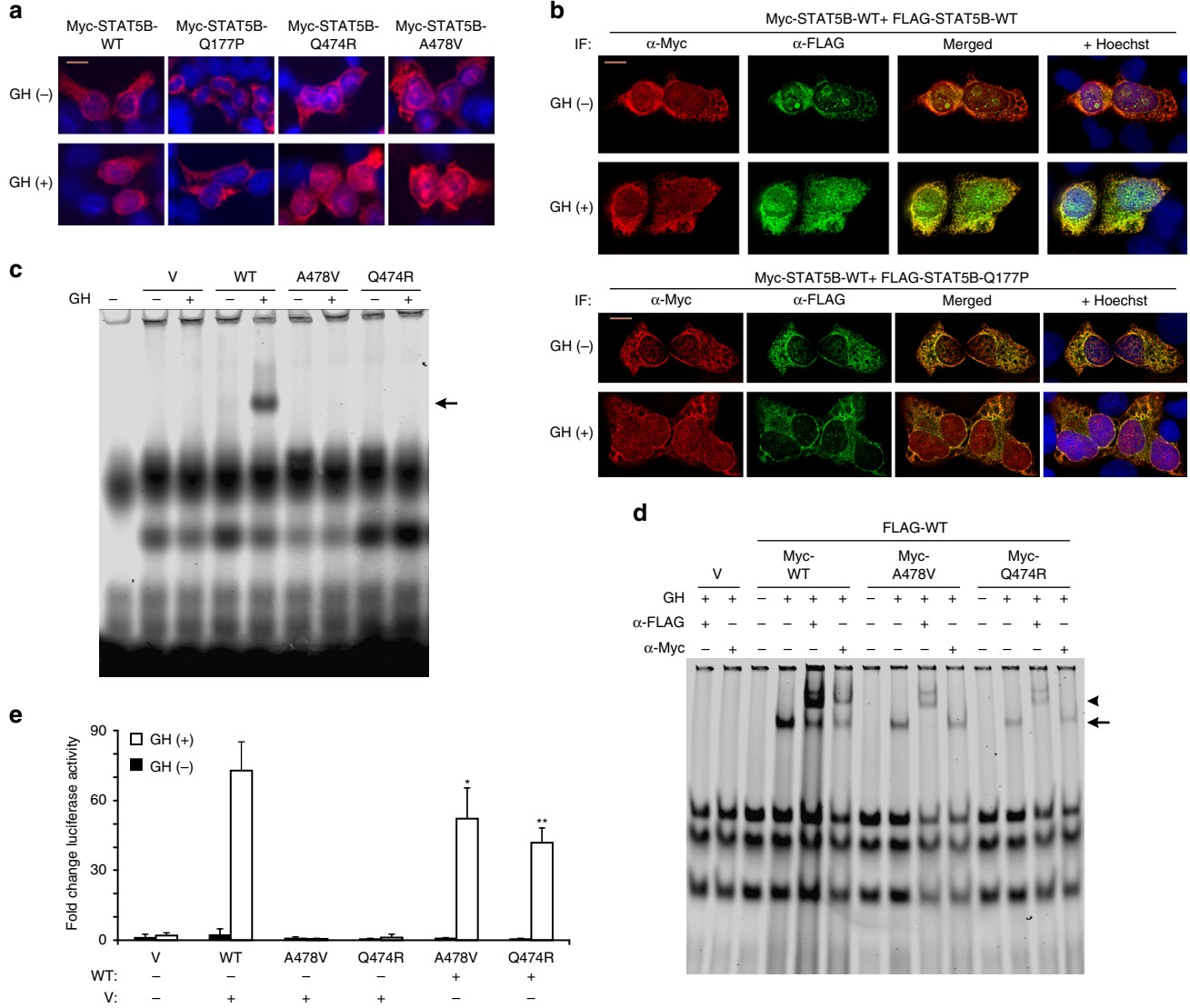

**Fig. 3** STAT5B missense variants are defective in nuclear translocation or DNA binding. **a** Representative immunofluorescence images of untreated or GH-stimulated HEK293(hGHR) cells transfected with Myc-STAT5B wild type (WT) or Myc-STAT5B variant plasmids (as indicated, abbreviations as in Fig. 2) and stained with α-Myc antibodies and Hoechst 33342 nuclear staining (scale bar, 10 μm). **b** Images of GH-stimulated or untreated HEK293(hGHR) cells transiently transfected with Myc-STAT5B wild type (WT) and co-transfected with plasmids expressing WT (upper panel) or p.Gln177Pro STAT5B (Q177P, bottom panel) followed by immunostaining (IF) with α-FLAG or α-Myc antibodies and Hoechst 33342 (scale bar, 10 μm). **c**, **d** Representative (three independent experiments) electrophoretic mobility shift assays (EMSA) (**c**) or supershift EMSA (**d**) to demonstrate DNA-binding of STAT5B WT and/or variant proteins as indicated. **d** Supershift EMSA of FLAG-WT co-transfected with indicated Myc-tagged WT or variants (V, vector only control; arrow, shifted; arrowhead, supershifted complexes). **e** Luciferase reporter activity measured in HEK293(hGHR) cells co-transfected with equimolar amounts of the indicated FLAG-tagged plasmids. Activities are presented as fold increase above unstimulated vector control, which was arbitrarily assigned a value of 1.0. Error bars represent mean ± S.D. of four independent experiments, each time in duplicates. *$p < 0.05$, **$p < 0.01$ relative to GH-stimulated WT, using one-way ANOVA with post hoc Tukey test

In Family 3, the father (I-1) also carried a private heterozygous *JAK2* variant *c.2374C > T* (p.Pro792Ser), which was transmitted to five of his six children, one of whom (II-4) did not carry the STAT5B p.Gln474Arg mutation. The binding of JAK2 by the homo-dimeric GHR, which, like all Type I cytokine receptor, lack intrinsic kinase activities, is crucial for activating the GH-induced GHR signaling cascades. Since II-4, age 7.9 years, was of normal stature (height SDS of −1.1), consistent with his normal IGF1 serum concentrations, we concluded that the JAK2 variant itself is unlikely to contribute significantly to postnatal growth retardation. With a sample size of only one, however, a potential synergistic effect with the STAT5B p.Gln474Arg mutation cannot be entirely ruled out. Subject II-4, furthermore, was non-syndromic (healthy, no evidence of eczema, normal IgE concentrations [Supplementary note 1]), suggesting that the JAK2 variant, by itself, did not contribute to the other symptoms observed in affected family members. To date, loss-of-function JAK2 mutations have not been reported while somatic and germline dominantly inherited gain-of-function JAK2 variants, particularly the recurrent p.Val617Phe, are well established causes of hematologic disorders (thrombocythemia-3, thrombocytopenia, polycythemia vera)[37–39]. In family 3, there was a lack of symptoms indicative of similar hematologic abnormalities, suggesting JAK2 p.Pro792Ser is not a dominant gain-of-function mutation. Altogether, the heterozygous JAK2 p.Pro792Ser is unlikely to be a main contributor towards the clinical symptoms

observed in family 3. It remains possible that the JAK2 variant is poorly penetrant in II-4 or is a recessive variant.

Our patients lacked the severe immune deficiency and pathological autoimmunity typically associated with total STAT5B deficiency. The immune complications of STAT5B deficiency are attributed to T-lymphopenia, most notably impairment in numbers and functions of T-regulatory cells (Treg) that express the transcription factor FOXP3, Treg[+]FOXP3[+], which are critical for maintenance of T-cell homeostasis[11,40]. Specifically, in STAT5B deficiency, it has been noted that peripheral, naïve Treg subsets were low, while dividing memory Treg subsets (CD45RO[+]Ki67[+]) were abnormally elevated and had reduced suppressive functionality, thus likely contributed to the autoimmune phenotypes of STAT5B-deficient patients[40,41]. Immunological profiles (B, NK, T-lymphocytes) of our present patients, in contrast, were unremarkable, with T-lymphocytes and subsets (Treg, Treg[+]FOXP3[+], naïve, and memory cells), in particular, to be within normal ranges (Supplementary Data 1).

An unexpected phenotypic feature in Family 3 was microcephaly in Proband 3 and two of her affected siblings but not in the carrier father. Whether the reduced head circumference was a consequence of the AD STAT5B mutation or, more likely, caused by additional genetic variants, remains unclear, but no obvious recessive or dominant candidate variants were revealed from available exome data. It is notable that the relatively normal head circumferences, albeit below the mean, of patients from families 1 and 2 were comparable to those observed in patients with GHIS[42] (Supplementary Fig. 3).

Additional major features frequently seen in AR STAT5B deficiency such as hyperprolactinemia[13] were detected in only a few of our AD STAT5B subjects (mother and aunt of family 2; Proband 3), and extrapulmonary autoimmune disease[41] was diagnosed only in Proband 3 (thyroiditis and celiac disease). Interestingly, elevated IgE concentrations manifested in only four of seven tested AR STAT5B-deficient patients[12,20,41] whereas eight of our nine AD STAT5B-deficient patients presented elevated IgE concentrations with the high penetrance, suggesting strong correlations between dominant actions of our STAT5B mutant proteins and IgE production. Mild eczema was reported for the majority of carriers of dominant-negative STAT5B mutations. Altogether, the clinical manifestations (i.e. growth impairment, elevated IgE, eczema) suggest variability in penetrance of the mutant genotypes and are consistent with the milder phenotype of AD STAT5B loss-of-function mutations (Supplementary Table 2 and Supplementary Fig. 2).

The prevalence of inactivating STAT5B mutations, either AR or AD, remains difficult to accurately determine from studies performed to date. Previous screenings for mutations or copy-number variants performed in short-statured patients (total $n = 377$), for example, did not reveal any pathogenic STAT5B variants[18,43–46], whereas in our present study (164 patients), three dominant-negative STAT5B mutations were identified. Future screenings of GHIS patients will validate whether the additional features of eczema and elevated IgE may be indications of potential inactivating AD STAT5B mutations. Intriguingly, each of the described loss-of-function germline STAT5B mutation is private, while reported somatic, heterozygous, STAT5B gain-of-function mutations associated with large granular lymphocytic leukemia or distinct types of T- or NK- lymphomas are clustered, often with the same mutation found in more than one individual[15–17,47–49]. The observation that STAT5B mutations, both germline loss-of-function and somatic gain-of-function, lead to clinical manifestations associated with dysfunctional T-lymphocytes, support critical roles of STAT5B in T-lymphocyte biology which cannot be compensated by the closely related STAT5A.

Both the clinical presentation of the patients and functional studies support residual normal STAT5B activities when in the heterozygous state. Since STAT5B actions require dimerization, we hypothesize that mutant and WT monomers captured in WT:mutant heterodimers are functionally inactive and only the remaining predicted 25% WT:WT homodimers are transcriptionally competent. While 25% of dimeric WT STAT5B is clearly not adequate for full GH responsiveness, cytokine signaling appears to be less sensitive to diminished concentrations of active STAT5B, thus explaining the absence of severe immunologic or pulmonary problems in our patients. Interestingly, in mouse models, severe growth restriction and immune deficiency were observed only in the knock-out of both the Stat5b and Stat5a genes, but expression of normal Stat5a/b as low as ~10% significantly reduced the severity of immunodeficiencies[50,51]. Finally, in our patients, the lack of severe immune problems and presence of residual WT STAT5B suggested that recombinant human GH or rhIGF1 therapy might be effective for normalizing growth, possibly dependent on the type of mutation. Effectiveness of either therapy or a combination of both treatments remains to be fully evaluated.

In conclusion, we report the first germline dominant-negative STAT5B mutations observed in patients who demonstrated milder but significant postnatal growth impairment, mild GH insensitivity, eczema, and elevated IgE. The STAT5B p.Gln177Pro variant is the first naturally occurring STAT mutation with a defect in nuclear localization, while the p.Gln474Arg and p.Ala478Val variants are DNA-binding deficient. These novel STAT5B mutations, acting through distinct pathophysiological mechanisms, manifested as milder clinical GHIS with general sparing of the immune system, thus broadening the clinical spectrum of STAT5B deficiency and GHI syndrome.

## Methods

**Patient consent and regulatory compliance**. Written informed consent to participate in the study was obtained from all adult subjects and parents of minors. The study protocol including skin biopsy sampling from control individuals was in accordance with the principles of the Declaration of Helsinki and was approved by the Ethical Review Committee of the University of Leipzig, the Institutional Regulatory Board of the Cincinnati Children's Hospital Medical Center, and the Institutional Regulatory Board of the UCL Great Ormond Street Hospital Institute of Child Health, London.

**WES and Sanger DNA Sequencing analyses**. Genomic DNA extracted from whole blood was used for Sanger dideoxy-sequencing and WES. All coding exons of the genes indicated in the "Detailed patient reports" (Supplementary Note 1), including STAT5B were PCR amplified and subjected to direct sequencing (ABI PRISM 3100 and 310 Genetic Analyzers; Thermo Fisher Scientific, Waltham, MA). Sequences were compared to public references (for accession numbers see Supplementary Table 5). Primer sequences can be provided on request.

WES was performed on genomic DNA samples from individuals indicated in Supplementary Note 1 ("Detailed Patient Reports"). One microgram of dsDNA was sheared by sonication to an average size of 200 base pair (bp) on a Covaris S2 instrument. Library construction was performed in an automated fashion on an IntegenX Apollo324. After nine cycles of PCR amplification using the Clonetech Advantage II kit, 1 μg of genomic library was recovered for exome enrichment using the NimbleGen EZ Exome V2 kit. Libraries were sequenced on an Illumina HiSeq2500, generating approximately 30 million paired end reads 125 bases long each. Reads were aligned to the human reference genome version 19 (GRCh37) with the Burrows Wheeler Aligner[52]. Our analysis methods utilized the Broad Institute's Genome Analysis Toolkit (GATK) and followed the pipeline described in DePristo et al.[53] along with the modifications listed in the "Best Practices" document on their website [http://www.broadinstitute.org/gatk/]. Stringent filtering was applied for rare, non-synonymous variants that passed quality filters and had a read depth of >10 reads. Further filtering was performed depending on the inheritance mode of short stature and GHIS characteristics in each family, specifically de novo dominant, homozygous recessive, compound heterozygous, or X-linked form of inheritance in family 1; in family 2, a dominant mode of inheritance from the mother's side of the family, X-linked recessive inheritance, homozygous autosomal recessive, or a compound heterozygous pattern as well as dominant inheritance; and dominant inheritance in family 3. For dominant modes of transmission, given the severity of the phenotype, we excluded all variants

present in public databases including the 1000 Genomes [http://www.1000genomes.org], Exome Aggregation Consortium [http://exac.broadinstitute.org/], and an internal exome database (>750 individuals; Cincinnati Children's Hospital Medical Center, Cincinnati, OH), for recessive inheritance models (homozygous or compound heterozygous) we excluded all variants that were present in these databases with a minor allele frequency above 0.005.

**Laboratory evaluation**. Endocrine and immunological evaluations were performed employing approved standard techniques established at the University Hospitals of Hradec Kralove (Czech Republic) and Leipzig (Germany) for family 1, Great Ormond Street Hospital (family 2), and the Mayo Medical Laboratories (Rochester, MN) and the Diagnostic Immunology Laboratory at Cincinnati Children's Hospital Medical Center (Cincinnati, OH) for family 3. Extended IGF1 generation tests were designed and performed to assess severity of GH resistance in family 2. The test followed a 3-step regimen, in which patients self-administered subcutaneous injections of rhGH at doses indicated in Supplementary Table 1, each step for 2 weeks with 4 weeks washout in between steps. Serum IGF1 and IGFBP3 were measured at start and end of each step using standard methods.

**Plasmids and site-specific mutagenesis**. *STAT5B* WT cDNA plasmid constructs (WT) with or without N-terminal FLAG-tag or Myc-tag (F-STAT5B and M-STAT5B, respectively)[54] were used as templates for generating the missense variants, p.Gln177Pro (*c.530A > C*), p.Gln474Arg (*c.1421A > G*), and p.Ala478Val (*c.1433C > T*) by site-specific mutagenesis (QuikChange Site-Directed Mutagenesis Kit; Stratagene, La Jolla, CA), following the manufacturer's protocol. The resultant cDNA constructs carrying the point mutations were verified by Sanger DNA sequencing.

**Cell culture and transfection**. Primary dermal fibroblasts were established from skin biopsies taken from *STAT5B* patients of family 1 (14.5 years) and healthy age/sex matched controls undergoing elective orthopedic surgeries. Fibroblasts were cultured in Dulbecco's modified Eagle's medium (DMEM) supplemented with 10% fetal-bovine serum (FBS), 100 U ml$^{-1}$ penicillin, and 100 µg ml$^{-1}$ streptomycin. Cells between passages 6 and 12 were seeded at a density of 100,000 cells ml$^{-1}$, grown to approximately 70% confluency before stimulation experiments. HEK293 cells (ATCC; LGC Standards, Wesel, Germany) and HEK293 stably transfected with the human GH receptor gene [HEK293(hGHR), kindly provided by Dr. Richard J. Ross (University of Sheffield, Sheffield, UK)][55] were cultured in DMEM supplemented with 10% FBS. Primary cells and cell lines were tested for mycoplasma contamination repeatedly. Cells were transiently transfected with pcDNA3.1 vector or plasmids encoding N-terminally FLAG- or Myc-tagged WT or variant STAT5B, using ExGene DNA (BIOMOL, Hamburg, Germany) or PolyJet In Vitro DNA (SignaGen, Rockville, MD) transfection systems, following the manufacturers' instructions. Briefly, 100,000 cells were seeded in six-well plates and cultured to 60–70% confluence before an ExGene or PolyJet/plasmid solution was added to cells (DMEM + 10% FBS). For stimulation experiments, cells were washed and serum-starved (DMEM with 0.1% bovine serum albumin) for 24 h prior to treatment with GH (100 ng ml$^{-1}$) or IFN-γ (100 ng ml$^{-1}$). Total cell lysates were collected 30 min post-treatment unless otherwise indicated.

**Western immunoblot and co-immunoprecipitation analysis**. Transfected HEK293(hGHR) cells were solubilized in lysis buffer (1 × phosphate-buffered saline (PBS), 1% Nonidet P-40, 0.5% sodium deoxycholate, 0.1% SDS, 10 mg ml$^{-1}$ phenylmethylsulfonyl fluoride, 100 nM sodium orthovanadate, and protease inhibitor mixture (cOmplete Mini-EDTA-free; Sigma, St. Louis, MO)), cell debris removed by centrifugation, and final whole-cell lysates stored at −20°C. For nuclear cellular extracts of transfected HEK293(hGHR) cells (six-well tissue culture plates), cells were trypsinized (0.25% trypsin) for 1 min, neutralized with DMEM/10% FBS, and cells collected in chilled Eppendorf tubes. Collected cells were washed in cold PBS and resuspended in 500 µl cold nuclear extraction buffer containing 1 mM sodium orthovanadate and protease inhibitor mixture. Nuclear extraction buffer contains 200 ml Hypotonic Buffer (25 mM Tris-HCl, pH 7.4, 10 mM NaCl and 7 mM MgCl$_2$), 1 mM EDTA, 1 mM EGTA, and 1 mM DTT. After 15 min incubation on ice, 25 µl of 10% Triton X-100 was added to lyse the cell membrane (5 min) The mixture was centrifuged at 14,000 rpm, 4°C, for 2 min, and the supernatant (cytosolic fraction) was stored at −80°C. The nuclear pellet was further washed in cold nuclear extraction buffer and centrifuged at 14,000 rpm, 2 min, 4°C. The washed nuclear pellet was resuspended in cold nuclear extraction buffer (with 1 mM sodium orthovanadate and protease inhibitor mixture) and stored at −80°C in 20 µl aliquots. For immunoblot analysis, 10 µg nuclear extract, 20 µg cytosolic extract, or 30 µg of whole-cell lysates were separated on 7% SDS-polyacrylamide gels, transferred to nitrocellulose membranes, and probed with anti-STAT5B (sc-1656, 1:1000; Santa Cruz, Dallas, TX), anti-FLAG M2 (F3165, 1:1000; Sigma, St. Louis, MO), anti-phospho-STAT5 (#9351, 1:1000 or #9359, 1:500; Cell Signaling, Danvers, MA), anti-Myc (#2278, 1:500; Cell Signaling), or anti-β-actin antibodies (A1978 or A5316, 1:5000; Sigma) as indicated. For immunoprecipitation prior to immunoblot analysis, tagged STAT5B variants were pulled down from 300 µg whole-cell lysate with anti-Myc sepharose beads (Cell Signaling). Secondary horseradish peroxidase-conjugated anti-mouse IgG or anti-

rabbit IgG antibodies, appropriate SuperSignal substrates (Thermo Fisher Scientific, Waltham, MA), and GE Healthcare's Hyperfilm ECL (Madison, WI) were used for chemiluminescence visualization. Uncropped captures of immunoblots are provided in Supplementary Figs. 4 and 5.

**Immunofluorescent microscopy**. HEK293(hGHR) cells, 5 × 10$^4$ per well, were seeded on poly-lysine-coated eight-well chamber slides (Thermo Fisher Scientific), and transfected with a total of 0.5 µg plasmid DNA. For co-transfections, a 1:1 ratio of plasmid DNA was employed. Following GH treatment (100 ng ml$^{-1}$, 30 min), cells were washed in cold PBS and fixed with cold methanol, 5 min at −20 °C. For immunofluorescent staining, cells were blocked with 5% goat serum and exposed overnight to primary mouse-anti-FLAG (1:500) and/or rabbit-anti-Myc (1:50) antibodies prior to treatment with secondary goat-anti-mouse-FITC (1:500; 626511; Thermo Fisher Scientific, Carlsbad, CA) and/or goat anti-rabbit-A555 antibodies (1:500; #A-21428; Life Technologies) for 4 h. Hoechst 33342 (Thermo Fisher Scientific) was added (15 min) and coverslips mounted with ProLong Gold (Thermo Fisher Scientific). Fluorescent images were obtained using a wide field deconvolution system (GE Healthcare) consisting of an inverted Nikon TE 200 Eclipse microscope, a Kodak CH350 CCD camera, and the Deltavision operating system. Images were acquired using a ×60 objective in a 1024 × 1024 format and deconvolved with nine iteration using SoftWoRx software (Applied Precision, GE Healthcare).

**Luciferase reporter assays**. HEK293(hGHR) cells (six-well plates) were transiently co-transfected[21] as described above with 1.0 µg pGL2: 8xGHRE (growth hormone response element) luciferase reporter plasmid and a total of 1.0 µg STAT5B WT and/or mutant cDNA plasmids or empty vector (each 0.5 µg) as indicated. Transfected cells were incubated overnight prior to an 8 h serum-free media treatment. Cells were stimulated with GH (100 ng ml$^{-1}$, 24 h) and cell lysates were collected and assayed using Promega's Luciferase Assay System (Promega, Madison, WI). Luciferase activity was measured on a Veritas Luminometer (Promega).

**Electrophoretic mobility shift assay**. 5′-Cy5.5-tagged rat spi2.1 GHRE (growth hormone response element; 5′-ACGCTTCTACTAATCCATGTTCTGAGAAA TCATCCAGTCTGCCCA-3′) was employed as probe for EMSAs[56,57]. Briefly, 50 fmol of the duplex DNA probe were incubated with 2 µg (standard EMSA) or 3 µg (supershift assays) nuclear extracts (see above) at room temperature for 20 min, protected from light, prior to size fractionation on a native 5% acrylamide gel (in standard tris-borate-EDTA buffer). Bands were visualized by scanning the gel on Li-Cor Odyssey CLx (Li-Cor Biotechnology, Lincoln, NE). For supershift EMSA, 2 µg of anti-FLAG antibody or 3 µg of anti-Myc antibody were included in the reaction mix and incubated at room temperature for an additional 10 min prior to size fractionation of the DNA–protein–antibody complexes.

**Quantitative reverse-transcriptase PCR amplification**. Total RNA was isolated using RNeasy plus Mini Kit (Qiagen, Hilden, Germany). A total of 100 ng of RNA was reverse transcribed using SuperScript III reverse transcriptase and random hexamer [p(dN)6] primers (Thermo Fisher Scientific). Steady-state mRNA expression was measured by quantitative real-time PCR using the 2× qPCR MasterMix Plus Low ROX (Eurogentec, Seraing, Belgium) with an ABI 7500 Sequence Detection System (Applied Biosystems, Darmstadt, Germany). Expression analysis for *STAT5B* was performed with TaqMan Gene Expression Assay (Hs00273500_m1; Applied Biosystems) and for *GHR* using primer/probe combination as described by Friedberg et al. (F: 5′-TTGGAATATTTGGGCTAACAG TGA-3′; R: 5′-CCTCCTCTAATTTTCCTTCCTTGAG-3′; P: 5′-AGGATTAA AATGCTGATTCTGCCCCCAG-3′)[58]. Transcript levels were normalized to the mean of two housekeeping genes using the following primer/probe combinations: β-Actin (*ACTB*): F: 5′-CGACGCGGCTACAGCTT-3′, R: 5′-CCTTAATGTCACG CACGATTT-3′, P: 5′-ACCACCACGGCCGAGCGG-3′; TATA-box-binding protein (*TBP*): F: 5′-TTGTAAACTTGACCTAAAGACCATTGC-3′, R: 5′-TTCG GTGGCTCTCTTATCCTCATG-3′, P: 5′-AACGCCGAATATAATCCCAAGCGG TTG-3′.

**Statistical analyses**. Data were analyzed by one-way ANOVA with post hoc Tukey test. Statistical analysis of the data was performed using GraphPad Prism 5 software (GraphPad Software, La Jolla, CA).

**Data availability**. The authors declare that all the data supporting the findings of this study are included in the article (or the Supplementary material) and available from the corresponding author (V.H.) upon reasonable request. Data of pathogenic mutations reported within this study have been deposited in ClinVar with the accession codes SCV000681436 (p.Gln177Pro), SCV000680478 (p.Gln474Arg), and SCV000693651 (p.Ala478Val). WES data that support the findings are not publicly available due to information that could compromise the research participants privacy/consent.

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

## Acknowledgements

The authors thank all participating families for their kind cooperation, Heike Pfäffle for assistance in targeted sequencing, and Kyle Buckham for technical assistance. This work was supported by funding from NIH NICHHD (R01HD078592 to V.H.), NIH NICHHD (1K23HD073351 to A.D.), and a Junior Research grant by the Medical Faculty of the University of Leipzig (to D.R.). M.T.D. receives funding from the Great Ormond Street Hospital Children's Charity (GOSHCC).

## Author contributions

S.F.A. and D.R. performed most of the experimental work. M.A.S. conducted immunofluorescence data acquisition and analysis. A.D. coordinated, performed and analyzed WES sequencing; J. Klammt, S.F.A., E.F.G., J. Kowalczyk, L.M., and V.H. coordinated, performed, and analyzed targeted sequencing. D.N., E.F.G., I.D.S., R.C., D.V., M.T.D., and R.P. collected clinical data and provided patient material. J. Klammt. and V.H. wrote the manuscript, E.F.G., R.G.R., and A.D. contributed to the manuscript. V.H. coordinated the project.

## Additional information

**Competing interests:** R.G.R. consults for OPKO, Versartis, Ascendis, Genexine, Ammonite, Sandoz, Ferring, and NovoNordisk. The remaining authors declare no competing interests.

