## [Peer Review File · Nature Communications]

Reviewers' comments:

Reviewer #1 (Remarks to the Author):

In this manuscript authors report 3 unrelated families which have novel germline STAT5b mutations associated with short stature and growth hormone insensitivity. In one family, the mutations seem to be de novo whereas in two other families also one of the parents is affected. By functional analyses authors show that mutated STAT5b can be efficiently phosphorylated upon stimulation, but in the first case mutated STAT5b is not able to enter in the nucleus. In two other mutation types nuclear entry is normal, but the mutated STAT5b is not able to bind DNA response elements. Authors conclude that mutations are inactivating and due to normally functioning wild-type STAT5b, the severity of the mutations is not as severe as in earlier reported loss of function mutations which truncate the whole STAT5b protein. Furthermore, authors conclude that the effect is mostly on the growth hormone axis and only minor immunological disturbances are observed. However, no detailed immunological analysis is presented in the paper.

Specific comments:

1. The frequency of novel STAT5b mutations as a cause of short stature would be important to analyze. Based on the manuscript, it is unclear how many families/cases have been studied in total. How were these 3 families picked up? Most probably the authors have also sequenced many other cases and therefore it would be important to report the whole study population. In addition, it would be good if authors could confirm their findings in a larger patient cohort with targeted STAT5b sequencing and in this manner discover how common/rare these mutations are in patients suffering from short stature.
2. In WES sequencing for family 3, 6 candidate mutations were found. These should be reported in the manuscript and some text should be added why these were excluded as possible causes. In addition, genes sequenced in targeted sequencing should be listed in supplemental tables or online methods.
3. Authors conclude in the abstract that mutations spared the immune system. However, many affected individuals had immunological disorders (asthma, celiac disease, infections) and therefore more thorough immunological evaluation is needed. STAT5b GOF mutations were first described in patients with CD8+ LGL lymphoproliferation concomitant with autoimmune disorders and more recently they have been described especially in cases with CD4+ proliferative disease. Therefore, it is likely that inactivating STAT5b mutations also affect immune cells although clinical manifestations can be mild. Therefore, at least basic immunophenotyping including different T cell subsets and memory classes should be analyzed from these patients and reported.
4. For the Gln177Pro mutation authors nicely show that it blocks the nuclear entry. Authors have also grown fibroblasts from the patient sample, but the nuclear visualization is only done with cell lines. Similar kind of stainings should be done with the patient sample. If material is no longer

available from fibroblasts, experiments can be performed with blood cells.

Reviewer #2 (Remarks to the Author):

In the present manuscript (NCOMMS-17-13498-T) Klammt et al. described familial cases of growth hormone insensitivity (GHI) in three families in which missense allelic variants in STAT5B gene segregated with the phenotype of GHI and eczema. The authors demonstrated by functional studies dominant negative effects of the identified missense variants in STAT5B, justifying the autosomal dominant inheritance of this phenotype.

This is the first description of autosomal dominant GHI caused by STAT5B mutations. Dominant negative variants were previously described for several other STATs proteins, and the present results were well contextualized. However, important points must be addressed by the authors.

1. Page 5, lines 82-86: Following the recommendation of ACMG/AMP (Richards et al. Genet Med. 2015 May 5;17(5):405–23), paternity and maternity should be confirmed for the twins with p.Gln177Pro STAT5B mutation.

2. Pages 12-13, lines 253-259. The 6 candidate variants obtained after exoma analysis should be showed in supplemental material.

3. Page S3, lines 104-106. The presence of microcephaly in three children from family 3 is intriguing. This was absent in the parents as well in the sibling with normal height who also carries a STAT5B variant. This could suggest an autosomal recessive inheritance model involving another gene. Did the authors perform exome sequencing analysis to investigate this possibility?

4. Table 1 and Table S2: Information about head circumference SDS should be included.

5. Page 10, lines 189-190: Could the authors hypothesize an explanation for intra and interfamilial variability of stature in the 3 families with autosomal dominant GHI caused by heterozygous STAT5B variants? Could the proportion between WT and Variant STAT5B be different in individuals with normal height (Family 2 II.2 and II.3; Family 3, II.6)? Could it be caused by monoallelic expression of the WT allele?

6. Page 10, lines 200-201: Since eczema is a frequent condition, the frequency of eczema in this limited number of carriers of dominant negative STAT5B variants should be analyzed in relation to its prevalence in the general population [Hanifin et al. Dermatitis. 2007 Jun;18(2):82-91].

Authors reply:

We thank the editor and reviewers for their constructive comments and helpful suggestions, which we have now addressed. As suggested, we have added the results of additional investigations, particularly whole-exome sequencing analysis for families 1 and 2, and more comprehensive immunological evaluations, to the revised version of the manuscript. These changes are referenced to the manuscript text in our point-by point reply below. Furthermore, some minor changes have been introduced to adapt the manuscript to format recommendations of the journal.

Reviewers' comments:

Reviewer #1 (Remarks to the Author):

In this manuscript authors report 3 unrelated families which have novel germline STAT5b mutations associated with short stature and growth hormone insensitivity. In one family, the mutations seem to be de novo whereas in two other families also one of the parents is affected. By functional analyses authors show that mutated STAT5b can be efficiently phosphorylated upon stimulation, but in the first case mutated STAT5b is not able to enter in the nucleus. In two other mutation types nuclear entry is normal, but the mutated STAT5b is not able to bind DNA response elements. Authors conclude that mutations are inactivating and due to normally functioning wild-type STAT5b, the severity of the mutations is not as severe as in earlier reported loss of function mutations which truncate the whole STAT5b protein. Furthermore, authors conclude that the effect is mostly on the growth hormone axis and only minor immunological disturbances are observed. However, no detailed immunological analysis is presented in the paper.

Specific comments:

- 1. The frequency of novel STAT5b mutations as a cause of short stature would be important to analyze. Based on the manuscript, it is unclear how many families/cases have been studied in total. How were these 3 families picked up? Most probably the authors have also sequenced many other cases and therefore it would be important to report the whole study population. In addition, it would be good if authors could confirm their findings in a larger patient cohort with targeted STAT5b sequencing and in this manner discover how common/rare these mutations are in patients suffering from short stature.*

Patients reported in the manuscript were identified as part of on-going, independent, research protocols focused on identifying molecular causes of GHIS associated short stature, performed at independent sites. For patients with IGF-I deficiency, targeted gene analysis along the GH-IGF-I axis is often performed.

The 3 cases reported in the manuscript were identified from a total of 164 independent, patients, corresponding to 1.83%. The total number of investigated cases is now added to the Results section (p 7, line 119) and Discussion (p 13, line 265). However, since the study population did not follow a traditional, formalized study protocol with predefined patient inclusion/exclusion criteria and was, in part, biased towards patients presenting with severe growth phenotypes, we felt that including a numerical frequency value may be misinterpreted, especially as there were several publications in which *STAT5B* variants were not identified in cohort studies (Discussion, p 13, last paragraph).

We agree with the reviewer that investigation of the *STAT5B* gene in a larger cohort would provide valuable insight into prevalence and, possibly, phenotypic heterogeneity of AD *STAT5B* deficiency. This will be done in the future, as such investigations would take quite some time to formalize and perform. Close attention can now also be paid to the additional phenotypic features identified in the present study. This point is now included in the Discussion section (p 14, lines 1-3).

2. *in WES sequencing for family 3, 6 candidate mutations were found. These should be reported in the manuscript and some text should be added why these were excluded as possible causes. In addition, genes sequenced in targeted sequencing should be listed in supplemental tables or online methods.*

We have now included tables that list the top candidates from WES analysis for all 3 families in the Supplementary material (Tables S3 and S4), including re-analysis of Family 3 using updated available genomic information (5 candidate variants), and discussed why only the *STAT5B* variants are likely causally linked to GHIS in the investigated family members – detailed WES analysis has been added in Supplementary material’s “Detailed patient reports” p S2, S4, S6. The descriptions of the initial targeted sequencing approach of families 1 and 2 were retained (Supplementary material’s “Detailed patient reports” p S2, line 53; pS4 line 122).

3. *Authors conclude in the abstract that mutations spared the immune system. However, many affected individuals had immunological disorders (asthma, celiac disease, infections) and therefore more thorough immunological evaluation is needed. *STAT5b* GOF mutations were first described in patients with CD8+ LGL lymphoproliferation concomitant with autoimmune disorders and more recently they have been described especially in cases with CD4+ proliferative disease. Therefore, it is likely that inactivating *STAT5b* mutations also affect immune cells although clinical manifestations can be mild. Therefore, at least basic immunophenotyping including different T cell subsets and memory classes should be analyzed from these patients and reported.*

Since our patients did not present clinical symptoms necessitating immunological profiling, immune evaluations were not performed as part of routine care. However, the point brought up

by the Reviewer is well taken, and immunological evaluations at certified clinical laboratories have now been performed, although not all tests were offered by the local laboratories. We have now included immunophenotyping of the majority of individuals who carry germline, dominant-negative STAT5B mutations, and confirmed that profiles (including T-regs, T-naïve and memory cells) were relatively normal, with the exception of IgE, which were elevated (Supplementary material, Table S5). Of note, T-reg⁺FOXP3⁺ cells could only be performed in Family 3, as such analysis were not available for Families 1 and 2 (only T-reg could be analyzed at respective sites). Discussions of profiling are now included in both Results section (p 7, second paragraph) and Discussion (p 12, second paragraph).

4. *For the Gln177Pro mutation authors nicely show that it blocks the nuclear entry. Authors have also grown fibroblasts from the patient sample, but the nuclear visualization is only done with cell lines. Similar kind of stainings should be done with the patient sample. If material is no longer available from fibroblasts, experiments can be performed with blood cells.*

We agree that similar experiments performed in primary cells would strengthen our hypothesis. Unfortunately, anti-STAT5B antibodies currently available cannot distinguish endogenous WT from Gln177Pro STAT5B, as the epitope recognized is located at the C-terminus of the STAT5B protein. Antibodies (various sources: G2 from Santa Cruz; anti-STAT5B from Boster; anti-STAT5B from Sigma) very poorly detected endogenous STAT5B in individual fibroblasts by IF and confocal, although STAT5B was readily detected in immunoblot analysis (reducing conditions, >30 ug cell lysates).

PBMCs, unfortunately, were not available. We tried treating primary fibroblasts with IFN-gamma, probing with phospho-Y-STAT5, IF and confocal visualization. As a positive control, we probed with pY-STAT1, which beautifully localized in the nucleus upon IFN-g stimulation (hardly any detection in the cytoplasm). Unfortunately, it was difficult to distinguish fluorescence of pY-STAT5 much above background, although concentrations of primary antibody, fixation methods, treatment times, were varied. Primary cells, incidentally, are poorly responsive to GH.

We consulted the Director of the Microscopy Facility (at Cincinnati Children's Hospital Medical Center), an expert in microscopy manipulation, and who, in the end, suggested genetic manipulations of primary cells, but agreed that this would "defeat" the purpose of determining what is happening to the endogenous proteins.

In summary, with good faith efforts, we attempted to address the concern of the Reviewer. Although not ideal, in our reconstitution studies, we took advantage of being able to construct N-terminally tags which permitted us to easily track WT and Gln177Pro STAT5B. We reasoned that if over-expressed Gln177Pro cannot mobilize into the nucleus upon stimulation, and prevent comparably expressed WT from translocating to the nucleus, then it is highly probable that in primary cells, when endogenous expression is low, a similar phenomenon is occurring.

Reviewer #2 (Remarks to the Author):

In the present manuscript (NCOMMS-17-13498-T) Klammt et al. described familial cases of growth hormone insensitivity (GHI) in three families in which missense allelic variants in STAT5B gene segregated with the phenotype of GHI and eczema The authors demonstrated by functional studies

dominant negative effects of the identified missense variants in STAT5B, justifying the autosomal dominant inheritance of this phenotype.

This is the first description of autosomal dominant GHI caused by STAT5B mutations. Dominant negative variants were previously described for several other STATs proteins, and the present results were well contextualized. However, important points must be addressed by the authors.

1. *Page 5, lines 82-86: Following the recommendation of ACMG/AMP (Richards et al. Genet Med. 2015 May 5;17(5):405–23), paternity and maternity should be confirmed for the twins with p.Gln177Pro STAT5B mutation.*

Whole exome sequencing – as requested by the editor - was performed in both affected twins as well as their parents and data were also used to confirm paternity and maternity. We have added the following to the Supplement (p S2, line 68): From WES data, the variant was further confirmed to be *de novo* by calculating the concordance rates between single nucleotide variants amongst the family members (Supplementary Table S7), which demonstrated that each affected child was confirmed to have ~50% concordance with each parent which is the expected rate for a parent-child dyad. Additionally, the siblings were confirmed to have >98% concordance rate for single nucleotide variants that pass filters, which further confirms that they are in fact identical twins. The small percentage of discordant calls is attributable to sequence coverage issues as well as sequencing artifacts.

2. *Pages 12-13, lines 253-259. The 6 candidate variants obtained after exoma analysis should be showed in supplemental material.*

see reviewer 1, comment 2

3. *Page S3, lines 104-106. The presence of microcephaly in three children from family 3 is intriguing. This was absent in the parents as well in the sibling with normal height who also carries a STAT5B variant. This could suggest an autosomal recessive inheritance model involving another gene. Did the authors perform exome sequencing analysis to investigate this possibility?*

We thank the reviewer for suggesting re-evaluating this interesting feature. First, to investigate whether there was a recessive gene contributing to microcephaly in family 3, we performed a secondary analysis of the exome data. Specifically, we considered that individuals II-1, II-2, and II-5 all were affected with microcephaly and the father and brother (II-3) who were whole-exome sequenced were not affected. There were no variants/genes that met analytic criteria for any one of the tested recessive inheritance patterns. Therefore, while we cannot conclusively rule out the possibility of an additional genetic variant contributing to the microcephaly seen in family 3, there are no clear recessive candidates seen in the available exome data. The analysis has now been included in Supplementary material (“Detailed patient reports”; p S6, last paragraph)

Secondly, we collected head circumference (HC) data for families 1 and 2 and also performed a review of the literature. As expected, there is a correlation between height and HC in the investigated (GHIS) conditions, summarized in Supplementary Fig. S3. In patients with classical

GHIS (Laron syndrome due to GHR mutations, AR STAT5B mutations) or isolated GH deficiency (IGHD; data as published by Laron, 2012¹) retardation of HC is less pronounced than the height deficit. In contrast, data from the few reported patients suffering from IGF deficiency caused by mutations in the *IGF1* gene have a severe, nearly “perfectly” proportionate HC to height ratio. A less severe but similar pattern can be observed for the carriers of the AD STAT5B mutation in family 3 (except the father). However, microcephaly is not a common feature of our AD STAT5B cases since carriers of families 1 and 2 presented with less reduced HC in line with the observations for classical GHIS and IGHD.

HC data and WES results are now summarized in the Results section (p 6, line 103) and included in Tables 1 and S2, discussed in the Discussion section (p 12, line 241) and presented in detail within the Supplementary material (“Detailed patient reports”; p S6, line 204) including the new Supplementary Fig. S3.

4. *Table 1 and Table S2: Information about head circumference SDS should be included.*

HC SDS was added to Table 1 and Supplementary Table S2.

5. *Page 10, lines 189-190: Could the authors hypothesize an explanation for intra and interfamilial variability of stature in the 3 families with autosomal dominant GHI caused by heterozygous STAT5B variants? Could the proportion between WT and Variant STAT5B be different in individuals with normal height (Family 2 II.2 and II.3; Family 3, II.6)? Could it be caused by monoallelic expression of the WT allele?*

We agree with the reviewer that variability of the major characteristics of any disease – in the present study, measurable as height variability – has always been perplexing, with lack of clear explanations. In the original version of the manuscript, Supplementary Figure S2 indicated, quite strikingly, that variability in stature in AD STAT5B deficiency (all carrier patients from present study) was comparable to GHIS due to dominant-negative GHR, and, further, was statistically different from AR STAT5B heterozygous carriers. We have added a corresponding sentence to the Discussion (p 12, line 224), indicating the range of height SDS associated with AD STAT5B.

Height variability in our patients may be caused by the same, genetic and non-genetic, factors that also determine individual growth in healthy subjects. Theoretically, carriers of dominant mutations might be particularly sensitive to modifying mechanisms such as random monoallelic expression (MAE) indicated by the reviewer. However, since most MAE occurs transiently² and MAE has not been reported to be of particular relevance for STAT5B expression (literature review and compare: <https://mae.hms.harvard.edu/genepage.php?geneID=4294>) we omitted speculation, albeit interesting, about its biological significance in the context of the present study.

6. *Page 10, lines 200-201: Since eczema is a frequent condition, the frequency of eczema in this limited number of carriers of dominant negative STAT5B variants should be analyzed in relation to its prevalence in the general population [Hanifin et al. *Dermatitis*. 2007 Jun;18(2):82-91].*

We agree with the reviewer that eczema frequency calculation in our patients compared to the general population would corroborate the suggested association of AD STAT5B mutations with the manifestation of this clinical feature. Because of the different group sizes of mutation carriers in this study (6/11; 63.6%) and the control population (12,424/116,202; 10.7%)³ we performed a 1-sample proportion test against the null hypothesis ($P_0=10.7\%$) that demonstrates the significant linkage between AD STAT5B mutations and eczema ($P < 0.0001$; 95% CI 23.38 to 83.25; note: similar, significant results are obtained if Pearson's chi-squared test or data by Shaw et al., 2011⁴ are applied). However, since our patient group is biased by partial genetic kinship and because it does not represent a random sample we are reluctant to include this information in the manuscript.

References

1. Laron, Z., Iluz, M. & Kauli, R. Head circumference in untreated and IGF-I treated patients with Laron syndrome: comparison with untreated and hGH-treated children with isolated growth hormone deficiency. *Growth Horm. IGF Res.* **22**, 49–52 (2012).
2. Reinius, B. et al. Analysis of allelic expression patterns in clonal somatic cells by single-cell RNA-seq. *Nat. Genet.* **48**, 1430-1435 (2016).
3. Hanifin, J.M. & Reed, M.L. A population-based survey of eczema prevalence in the United States. *Dermatitis* **18**, 82–91 (2007).
4. Shaw, T.E., Currie, G.P., Koudelka, C.W. & Simpson, E.L. Eczema prevalence in the United States: data from the 2003 National Survey of Children's Health. *J. Invest. Dermatol.* **131**, 67–73 (2011).

Reviewer's comments

Reviewer #1 (Remarks to the Author):

Authors have revised the manuscript according to suggestions raised by the reviewers and editors. They have performed basic immunological analyses from patients and no severe alterations were found except high IgE levels. In addition, they have performed additional WES analysis and report that data. In the supplementary file, they report the other candidate variants. For family 3, it is quite intriguing that also JAK2 mutation was observed. Authors exclude this mutation as it is present also in one brother with no severe growth retardation, but actually also some growth delay is also observed in him. IgE is normal, but that is also normal in one affected sibling.

In the table they present, this mutation is predicted to be damaging. Has this mutation been reported earlier? Is it predicted to be loss or gain of function? Could that play additional role here? Authors could add this candidate mutation in the actual text as it is clearly affecting the same pathway and based on their data it cannot be ruled out that it could have some additional effect.

Reviewer #2 (Remarks to the Author):

The authors have satisfactorily addressed my concerns.

Authors Response:

We thank the editor and Reviewer #1 for the opportunity to address additional concerns.

Reviewers' comments:

Reviewer #1 (Remarks to the Author):

Authors have revised the manuscript according to suggestions raised by the reviewers and editors. They have performed basic immunological analyses from patients and no severe alterations were found except high IgE levels. In addition, they have performed additional WES analysis and report that data. In the supplementary file, they report the other candidate variants. For family 3, it is quite intriguing that also JAK2 mutation was observed. Authors exclude this mutation as it is present also in one brother with no severe growth retardation, but actually also some growth delay is also observed in him. IgE is normal, but that is also normal in one affected sibling. In the table they present, this mutation is predicted to be damaging. Has this mutation been reported earlier? Is it predicted to be loss or gain of function? Could that play additional role here? Authors could add this candidate mutation in the actual text as it is clearly affecting the same pathway and based on their data it cannot be ruled out that it could have some additional effect.

We agree with the Reviewer that the novel JAK2 variant, c.2374C>T, p.Pro792Ser, could be more thoroughly discussed. All heterozygous JAK2 mutations described to date have been associated with blood disorders, which was clearly absent in the one brother (II-4) who carried the JAK2 variant and was wild-type for STAT5B, and also absent in the affected family members. II-4, in addition, had normal height at age 7.9yr. We have included relevant statements in Results, Supplementary Information ("Detailed patient reports"), and have devoted a paragraph in the Discussion on this topic (page 13):

"In Family 3, the father (I-1) also carried a private heterozygous *JAK2* variant c.2374C>T, (p.Pro792Ser) which was transmitted to 5 of his 6 children, one of whom (II-4) did not carry the STAT5B p.Gln474Arg mutation. The binding of JAK2 by the homo-dimeric GHR, which, like all Type I cytokine receptor, lack intrinsic kinase activities, is crucial for activating the GH-induced GHR signaling cascades. Since II-4, age 7.9yr, was of normal stature (height SDS of -1.1), consistent with his normal serum IGF-I concentrations, we concluded that the JAK2 variant itself is unlikely to contribute significantly to post-natal growth retardation. With a sample size of only one, however, a potential synergistic effect with the STAT5B p.Gln474Arg mutation cannot be entirely ruled out. Subject II-4, furthermore, was non-syndromic (healthy, no evidence of eczema, normal IgE concentrations; Supplementary Information "Detailed patient reports"), suggesting that the JAK2 variant, by itself, did not contribute to the other symptoms observed in affected family members. To date, loss-of-function JAK2 mutations have not been reported while somatic and germline dominantly inherited gain-of-function JAK2 variants, particularly the recurrent p.Val617Phe, are well established causes of hematologic disorders (thrombocytopenia-3, thrombocytopenia, polycythemia vera). In family 3, there was a lack of symptoms indicative of similar hematologic abnormalities, suggesting JAK2 p.Pro792Ser is not a dominant gain-of-function mutation. Altogether, the heterozygous JAK2 p.Pro792Ser is unlikely to be a main contributor towards the clinical symptoms observed in family 3. It remains possible that the JAK2 variant is poorly penetrant in II-4 or is a recessive variant."